# Foam Rolling of the Calf and Anterior Thigh: Biomechanical Loads and Acute Effects on Vertical Jump Height and Muscle Stiffness

**DOI:** 10.3390/sports7010027

**Published:** 2019-01-19

**Authors:** Christian Baumgart, Jürgen Freiwald, Matthias Kühnemann, Thilo Hotfiel, Moritz Hüttel, Matthias Wilhelm Hoppe

**Affiliations:** 1Department of Movement and Training Science, University of Wuppertal, Fuhlrottstraße 10, 42119 Wuppertal, Germany; freiwald@uni-wuppertal.de (J.F.); matthiaskuehnemann@gmx.de (M.K.); m.hoppe@uni-wuppertal.de (M.W.H.); 2Department of Orthopedic, Trauma, Hand and Neuro Surgery, Klinikum Osnabrück GmbH, Am Finkenhügel 1, 49076 Osnabrück, Germany; thilo.hotfiel@klinikum-os.de; 3Department of Orthopedic Surgery, Friedrich-Alexander-University Erlangen-Nürnberg, Rathsberger Straße 57, 91054 Erlangen, Germany; moritz.huettel@gmx.de

**Keywords:** warm-up, force, cycling, muscle, self-myofascial release

## Abstract

When considering the scientific lack concerning the execution and acute effects and mechanism of foam rolling (FR), this study has evaluated the biomechanical loads by the force-time characteristics during two popular FR exercises. Additionally, the acute effects of FR on jump height and muscular stiffness were simultaneously assessed. Within a randomized cross-over design, 20 males (26.6 ± 2.7 years; 181.6 ± 6.8 cm; 80.4 ± 9.1 kg) were tested on different days pre, post, and 15 and 30 min after three interventions. The interventions consisted of a FR procedure for the calf and anterior thigh of both legs, 10 min ergometer cycling, and resting as a control. Stiffness was measured via mechanomyography at the thigh, calf, and ankle. The vertical ground reaction forces were measured under the roller device during FR as well as to estimate jump height. Within the FR exercises, the forces decreased from the proximal to distal position, and were in mean 34 and 32% of body weight for the calves and thighs, respectively. Importantly, with 51 to 55%, the maxima of the individual mean forces were considerably higher. Jump height did not change after FR, but increased after cycling. Moreover, stiffness of the thigh decreased after FR and increased after cycling.

## 1. Introduction

In recent years, the popularity of foam rolling (FR) has increased. Its postulated benefits are to optimize warm-up procedures, to enhance performance and recovery, and also to treat fascial disorders [1,2,3]. However, despite its popularity, the effects of FR are not well investigated. This is especially true concerning performance alterations [4,5]. Consequently, more studies to investigate the effects of FR on sportive performances are needed. A major problem of the investigation and application of FR is that the biomechanical load has been poorly studied [4]. The few existing studies have measured the force passing through a roller device using a force plate and revealed mean forces between 29 and 50% of body weight during FR of the anterior thigh [6,7]. Additionally, the applied forces differ between proximal and distal positions during FR of the posterior thigh [8]. To standardize the biomechanical load, some studies used constant loads to investigate the effects of FR [9,10]. Interestingly, one study has compared range of movement enhancements after FR, depending on high and low subjective selected loads, and found no differences [11]. However, no study has evaluated the biomechanical loads during different FR exercises, which are frequently conducted in practice yet.

The ability to generate high muscular force per time, such as during vertical jumps, is an important physical factor in many sports [12]. Concerning the impact of FR, inconsistent findings of the acute effects on jump performance, which were potentially based on different assessment protocols and/or performance levels, hinder a generalization. In fact, immediately after FR, jump performance was decreased [13], unchanged [11,14,15], or enhanced [16]. Despite these inconsistent effects of FR on jump performance, the underlying mechanisms are also unclear. Thus, more research to investigate the effects of FR on vertical jump performance and potential underlying mechanisms is required. One factor, which may explain alterations in jump performance after FR, is related to changes in muscular stiffness [16,17]. After FR, two studies have shown a decrease in the stiffness of anterior and posterior thigh muscles [18,19]. Additionally, one study found a decrease in the M. vastus lateralis, but not in the M. rectus femoris [6]. A further study has found no change in M. rectus femoris of the treated leg, but a decrease in the contralateral leg [20]. Thus, more research is needed to understand the effects of FR on changes in muscular stiffness, as one potential factor that impacts vertical jump performance.

This study aimed (1) to evaluate the biomechanical loads during different FR exercises and (2) to investigate the acute effects on vertical jump height and tissue stiffness.

## 2. Materials and Methods

Twenty recreational male athletes (age: 26.6 ± 2.7 years; body height: 181.6 ± 6.8 cm; body mass: 80.4 ± 9.1 kg) participated. All of the athletes were free of acute or chronic musculoskeletal, neurological, and cardiovascular disabilities. The athletes had no experience with the application of FR. To ensure a proper execution, the athletes were instructed in the investigated FR exercises one week before the first measurement. They were also instructed to continue their regular sportive activities during the study. After the explanation of all testing procedures, the athletes gave their written consent to participate. The study was accepted by the Ethics Committee of the local university without revisions (150521 MS/BB).

### 2.1. Research Design

The research design of our study is summarized in Figure 1. To evaluate the biomechanical loads of two frequently conducted FR exercises, vertical ground reaction forces during rolling were measured. A randomized cross-over design was used to investigate the effects of FR on vertical jump height and muscular stiffness. Therein, the effects of FR were compared with those of ergometer cycling (CYC) and a control condition (CON). The CYC was used as a further intervention, because several studies have shown that warm-up procedures, involving ergometer cycling, enhance vertical jump performance [21,22]. This may allow for a better evaluation of the practical relevance of potentially effects after FR. The vertical jump height and muscular stiffness were assessed at the following time points: pre, post, and 15 min, as well as 30 min after each intervention. All of the interventions were performed over two weeks, with one week between each.

### 2.2. Intervention Procedures

The FR exercises were performed on the anterior thigh and calf of both legs with a conventional high-density foam roller device (length 30 cm, diameter 15 cm; BLACKROLL^®^, Bottighofen, Switzerland). For the anterior thighs, the athletes were in a plank position. The treated leg was placed on the foam roller device, while the foot of the non-treated leg had contact with the ground. The rolling was performed between the top of the patella and the anterior superior iliac spine. The FR of the calves was performed in an adopted seated position with the hands keeping the body off the ground and the non-treated leg being crossed above the treated one. The exercise was conducted between the popliteal fossa up and the myotendinous junction of the Achilles tendon. All of the athletes completed two sets of 30 repetitions for the anterior thighs and calves. The conducted FR exercises are presented in Figure 1. The CYC consists of a 10 min cycling on a stationary ergometer (Cyclus2, RBM, Leipzig, Germany) at a moderate intensity. The intensity was guided by the Borg scale using a rated perceived exertion value of 12–14 of a maximum of 20 [23]. The corresponding mean power output was 137 ± 24 W. During CON, and between all post-measurements, the athletes rested in supine position on a massage bench.

### 2.3. Output Measures

During the FR exercises, the vertical ground reaction forces were measured at 300 Hz using a force plate (Type 9287BA, Kistler, Winterthur, Switzerland). The center of pressure was analyzed to identify the repetitions and movement directions. After time normalization, a mean force curve was calculated in percent of the body weight for each FR exercise. The force curves were then averaged for both legs.

To estimate changes in vertical jump height, the athletes performed three counter movement jumps without arm swing. During the jumps, vertical ground reaction forces were sampled at 1000 Hz using the force plate specified before. The vertical jump height was calculated by the impulse-momentum method [24]. To increase the reliability, the average jump height was used for statistical analyses.

To measure muscular stiffness a myomechanographic device (MyotonPRO, Myoton AS, Tallinn, Estonia) was used. This device measures the mechanical oscillation of the tissue that was provoked by a defined mechanical impact. After applying a vertical preload of 0.18 N to the skin, five short (15 ms) mechanical impacts of 0.4 N were applied to the tissue. The tissue response was then measured by an acceleration sensor. Here, stiffness was defined as the resistance of the tissue to the applied force. On both legs, two points at the thigh and calf were selected to take the effects of FR on muscular stiffness into account. A third point at the ankle was selected as control condition, where no muscle is located. At the thigh, the device was placed over the M. rectus femoris at 50% of the distance between the anterior spina iliaca superior and the superior part of the patella. The measurement was executed in the supine position. At the calf, the device was placed over the M. gastrocnemius medialis at the maximum circumference. At the ankle, the point was defined in the middle of the horizontal line between the lateral malleolus and Achilles tendon. For the latter two measurements, the athletes rested in prone position with the feet hanging above the massage bench to achieve an ankle joint angle of 90°. To increase the reliability, the stiffness values of both legs were averaged for all points. On the first day, the three points were marked with a water-resistant marker for accurate reproducibility during the study. A sufficient test-retest reliability (ICC ≥ 0.80) of the myomechanographic device has been reported for the M. rectus femoris and M. gastrocnemius medialis before [25,26].

### 2.4. Statistical Analyses

All of the statistical calculations were executed with the SPSS 24.0 software package (IBM, Armonk, NY, USA). A significance level of *p* < 0.05 was used for all computations. After checking for normal distribution using Shapiro-Wilk tests, baseline (pre) differences between the interventions were evaluated using one-way repeated measure ANOVAs. Subsequently, two-way repeated measure ANOVA’s (time x intervention) were applied to the relative baseline differences at post, and 15 min and 30 min after each intervention. Moreover, one-way repeated measure ANOVA’s with Bonferroni post hoc tests were used to assess changes within and between the interventions. For the repeated ANOVA’s, the assumption of sphericity was checked with the Mauchly’s test, and if required the degrees of freedom were adjusted using Greenhouse–Geisser correction. Effect sizes were calculated using partial eta-squared (η_P_²), with ≥0.01 indicating small, ≥0.059 medium, and ≥0.138 large effects [27].

## 3. Results

The mean vertical ground reaction force curves during the FR exercises are shown in Figure 2. Generally, the progression of the force curves were similar for the thigh and calf. The highest force values were present at the proximal position. The load decreased, when the roller device was moved distally. A mean force of 34 and 32% of the body weight were registered for the thigh and calf, respectively. The corresponding maxima of the individual force curves of the patients were 55 and 51%, respectively.

All mean vertical jump height and stiffness values are shown in Table 1. No significant baseline differences in vertical jump height (*p* = 0.727; η_P_² = 0.02) and stiffness of the thigh (*p* = 0.991; η_P_² < 0.01), calf (*p* = 0.481; η_P_² = 0.03), and ankle (*p* = 0.920; η_P_² < 0.01) were found between the three interventions. For each parameter, the relative changes and results of the two-way repeated measure ANOVA (time x intervention) are presented in Figure 3. Significant interaction effects (time x intervention) were found for the vertical jump height (*p* < 0.001) and stiffness of the thigh (*p* < 0.001), while single time-effects were present for the stiffness of the calf (*p* < 0.001) and ankle (*p* < 0.001).

After FR, the vertical jump height did not change, while it significantly increased after CYC and decreased after CON. At post and post 15 min, the vertical jump height after CYC was significantly higher than after CON and FR, as well as than FR at post 30 min (Figure 3a). After all interventions, the vertical jump heights were significantly lower at post 30 min as compared to the pre values.

In thigh stiffness, significant differences in the post values were found between all three interventions. While the stiffness of the thigh significantly decreased after FR, it increased after CYC, with no changes after CON. At post 15 and 30 min, no significant differences as compared to the baseline values as well as between the interventions were found. Regarding the stiffness of the calf and ankle, no significant differences between the interventions were found at any time point. However, an overall increase in calf and ankle stiffness appeared from post to post 30 min, as all values between these time points differed significantly, with exception of the calf-values during FR.

## 4. Discussion

This study aimed (1) to evaluate the biomechanical loads during different FR exercises and (2) to investigate the acute effects on vertical jump height and tissue stiffness. Our major outcomes were: (1) During the FR exercises, the mean forces applied to the thigh and calf were 34 and 32% of the body weight, respectively. (2) After FR, the vertical jump height did not change, whereas the muscular stiffness of the thigh decreased.

### 4.1. Biomechanical Load

The present study evaluated the biomechanical load during FR by a quantification of the vertical ground reaction forces. Mean forces of 34 and 32% of the body weight were measured, while threating the thighs and calves, respectively. Importantly, the maxima of the individually mean forces were clearly higher (thigh: 55% BW; calf: 51% BW). To date, no study has measured the biomechanical load during the same FR exercises that are frequently conducted in practice. In one study, the participants performed FR exercises of the thigh by lifting the non-treated leg off the ground [7]. The reported mean force of 50% body weight (range 27–67%) was therefore higher than in our study. However, our results are plausible, as, during the simultaneously FR of both anterior thighs, the mean forces (59–69% BW) were about twice as high as our values [6]. During both FR exercises, the ground reaction forces varied similarly from distal to proximal independently of the movement direction. The distal force values were about 15 to 20% of body weight lower than the proximal values. Similar force differences (21% BW) were shown between distal and proximal body positions of a FR exercise treating the posterior thigh [8], which supports our outcomes.

The amount and variation of the external biomechanical loads during FR have to be considered in future studies, as some studies have used constant and low (13 kg and 25% of the body weight) loads during FR exercises [9,10]. Moreover, also during training and therapy procedures, the external biomechanical loads should be quantified in the future. Thereon, the challenge will be the estimation of the internal loads during FR, as the same external load can lead to different pressure between the roller device and underlying tissue according to the roller type and also to different anatomical characteristics of a person [28]. During FR, the pressure could twice exceed the pressure that is used in occlusion studies [28,29]. Therefore, high mechanical compressions to the underlying tissue are induced, which can lead the harmful effects on the connective tissue, nerves, vessels, and bones, which requires further research [5]. For a safe and effective application of FR in therapy and training procedures, guidelines have to be developed, which account for the biomechanical load and include dose-response relationships.

### 4.2. Vertical jump Height

After FR, the vertical jump height did not change. This finding supports the results of previous studies [11,14,15]. Across all time points, between FR and CON, no differences were found. Therefore, FR as a stand-alone procedure seems not beneficial for increasing jump performance, but also no adverse effects were obtained. Further research is warranted to identify the influence of other variables that are involved in FR application on performance outcomes (e.g., biomechanical load, duration, sets), as well as its potentially long-term effects.

In contrast to FR, the jump height increases immediately after CYC by 4.6%. Therefore, a classical non-specific warm-up procedure is more beneficial to increase vertical jump performance [21,22]. Also at post 15 and 30 min, the jump height after CYC was higher than that after FR and/or CON, while a general decrease was present, when compared to the pre-values. Consequently, the positive effects of 10 min CYC on the vertical jump height were declined after 15 min.

### 4.3. Tissue Stiffness

The absolute stiffness values of the thigh and calf were comparable with previous studies that have used similar methods [17,25,30]. However, the revealed stiffness values at the ankle were considerably higher, which has not been reported so far. The stiffness of the thigh was differently influenced by the three applied interventions, while no differences were found at the calf and ankle. At the thigh, the stiffness decreased immediately after FR, increased after CYC, and remained unchanged in CON. The decrease after FR is in line with previous results [6,18,19], but it contradicts those of another study [20]. Also after a classical massage, a decrease in tissue stiffness was reported [31]. Two possible mechanisms for the reduction of the stiffness can be suspected, which are the breaking of resting cross-bridges and an increase in intramuscular temperature [31]. However, the decrease in stiffness has to be seen as a short-term effect as the values returned to baseline within 15 min, which is also in accordance with previous results [18]. Interestingly, in one study, the decrease in anterior thigh stiffness was independent of the FR speed [19]. In contrast, the increase in stiffness after CYC is explainable by the increased blood flow [32]. However, even after CYC, the change in thigh stiffness returns to the baseline within 15 min.

At the calf, no changes in the stiffness values were found after the interventions. Regarding FR, it can be mentioned that different effects on tissue stiffness can be caused by various exercises and/or treated muscles. Also after CYC, no changes in tissue stiffness were found at the calf. However, it has to be considered that CYC was performed with the foot in a posterior pedal position, which is less demanding for calf muscles [33]. Therefore, the warm-up effects of CYC in the calf muscles were potentially lower than those at the thigh.

Overall, single time effects in the stiffness values were found for the calf and ankle, but there were no differences between the interventions. The stiffness values increased stepwise from post to post 30 min. However, we have no clear explanation for this increase. One explanation may be a temperature effect. As the stiffness increased during cryotherapy [34], the skin and/or muscle temperature potentially decreased during resting, which was not controlled for. It is mentionable that muscle effects were unlikely as at the ankle no muscle is located.

### 4.4. Study Limitations

While our study revealed new practical relevant knowledge, few limitations are worth mentioning. By using mechanomyography to investigate changes in muscular stiffness, it is not possible to separate the effects of FR on different types of tissues (e.g., skin, fat, connective tissue). However, we have tried to account for that drawback by the selection of the different anatomical locations with different amounts of underlying tissues. A further limitation is that we only have investigated short-term effects and focused on changes in passive stiffness. Thus, potential long-term effects of FR, also with respect to changes in active stiffness during stretch-shortening, remain unknown, for which more research is needed.

## 5. Conclusions

This study shows that the mean forces applied to the thigh and calf were 34 and 32% of the body weight during two popular FR exercises. Additionally, the vertical jump height did not change after the FR exercises, whereas the muscular stiffness of the thigh decreased immediately after FR and then returned to baseline within 15 min.

## Figures and Tables

**Figure 1 sports-07-00027-f001:**
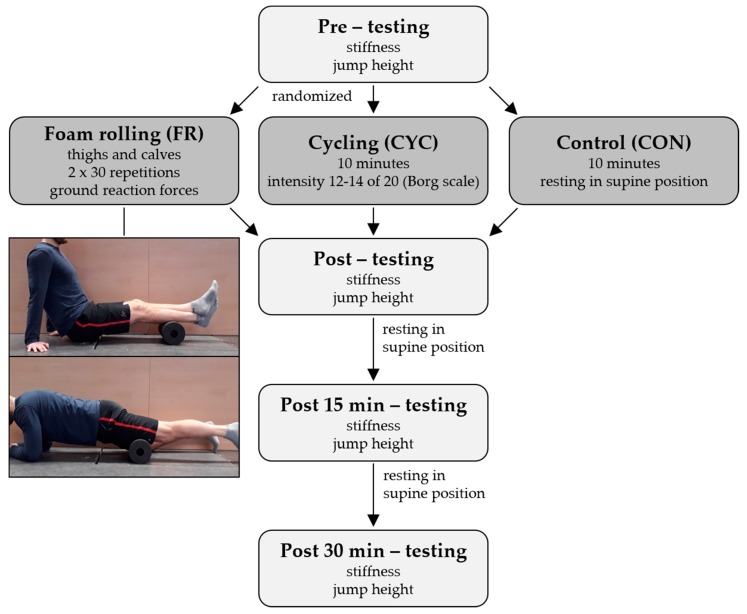
Research design.

**Figure 2 sports-07-00027-f002:**
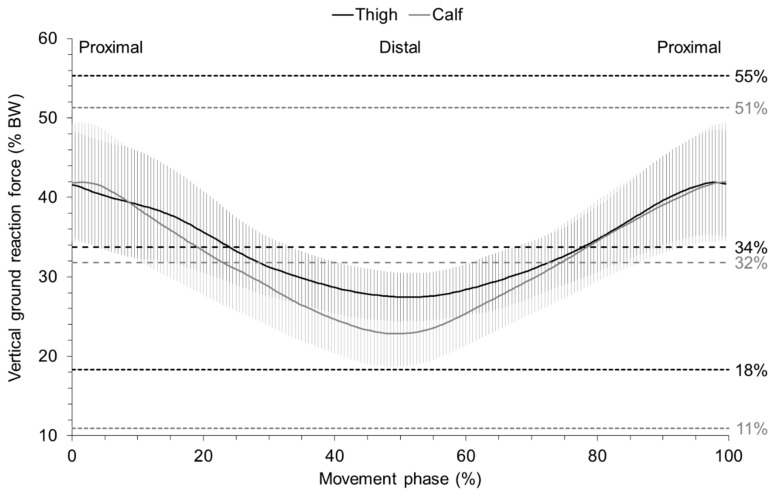
Relative vertical ground reaction forces measured during foam rolling for the anterior thigh and calf Notes: mean curves (solid lines); 90% CI (shaded area); overall means, maximum, and minimum values (dotted lines), BW (body weight).

**Figure 3 sports-07-00027-f003:**
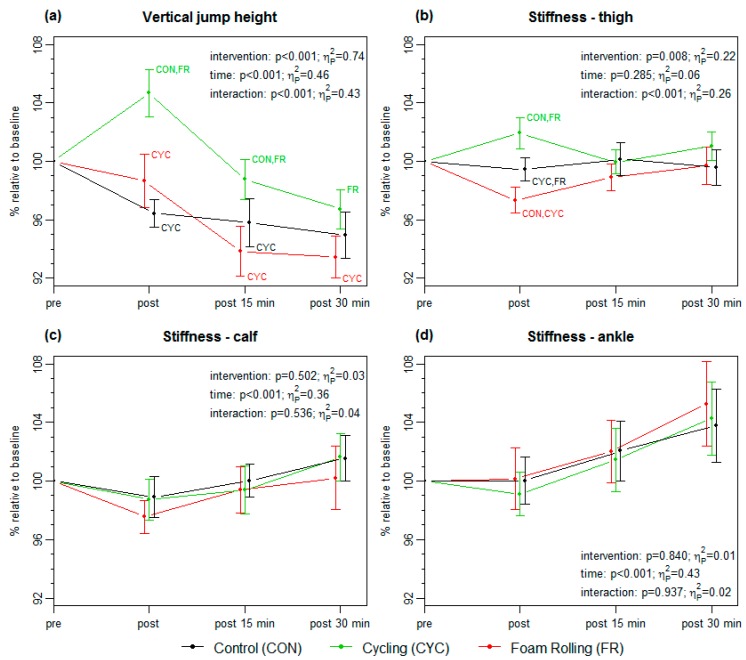
Percentage changes (mean ± 90% CI) in vertical jump height (**a**) and stiffness measurements at the thigh (**b**), calf (**c**), and ankle (**d**). The results of the two-way repeated measure ANOVAs (time x intervention) are shown within each plot. Within intervention comparisons using one-way repeated measure ANOVAs with Bonferroni post hoc tests revealing significant differences in vertical jump height (post *p* < 0.001, η_P_² = 0.58; post 15 min *p* < 0.001, η_P_² = 0.33; post 30 min *p* = 0.005; η_P_² = 0. 25) and thigh stiffness (post *p* < 0.001, η_P_² = 0.45).

**Table 1 sports-07-00027-t001:** Absolute stiffness and vertical jump height values (mean ± 90% CI) pre, post, post 15 min, and post 30 min after the foam rolling (FR), cycling (CYC), and control (CON) procedures.

Parameter		Inter-Vention	Pre	Post	Post 15 min	Post 30 min	ANOVA
Vertical jump height (cm)	Counter movement jump	FR	34.6 ± 1.0^3.4^	34.1 ± 1.2^3.4^	32.4 ± 1.1^1.2^	32.3 ± 1.1^1.2^	*p* < 0.001; η_P_² = 0.63
		CYC	34.8 ± 1.1^2.4^	36.4 ± 1.0^1.3.4^	34.4 ± 1.1^2.4^	33.7 ± 1.1^1.2.3^	*p* < 0.001; η_P_² = 0.70
		CON	34.7 ± 1.3^2.3.4^	33.4 ± 1.3^1^	33.2 ± 1.3^1^	32.9 ± 1.3^1^	*p* < 0.001; η_P_² = 0.46
Stiffness (N/s)	Thigh	FR	267.1 ± 10.5^2^	260.1 ± 10.9^1.3.4^	264.2 ± 11.0^2^	266.2 ± 10.7^2^	*p* < 0.001; η_P_² = 0.31
		CYC	267.4 ± 11.1^2^	272.7 ± 11.8^1.3^	267.1 ± 10.6^2^	270.0 ± 10.7	*p* = 0.003; η_P_² = 0.22
		CON	267.5 ± 11.7	266.0 ± 12.1	268.2 ± 13.1	266.8 ± 13.2	*p* = 0.426; η_P_² = 0.04
	Calf	FR	266.7 ± 12.5	259.7 ± 10.1	264.0 ± 8.1	265.9 ± 8.0	*p* = 0.113; η_P_² = 0.12
		CYC	265.3 ± 9.9	261.5 ± 8.1^4^	263.0 ± 7.6^4^	268.9 ± 7.4^2.3^	*p* = 0.019; η_P_² = 0.19
		CON	261.9 ± 8.7	258.8 ± 8.2^4^	261.7 ± 8.1	265.7 ± 8.7^2^	*p* = 0.017; η_P_² = 0.19
	Ankle	FR	549.1 ± 31.2^4^	548.9 ± 31.2^4^	558.5 ± 28.7	574.9 ± 27.3^1.2^	*p* = 0.001; η_P_² = 0.24
		CYC	553.4 ± 24.9^4^	548.0 ± 24.9^4^	560.8 ± 25.7	575.9 ± 25.8^1.2^	*p* < 0.001; η_P_² = 0.28
		CON	550.7 ± 33.7	550.9 ± 35.9^4^	562.2 ± 36.3	570.7 ± 35.3^2^	*p* = 0.002; η_P_² = 0.22

Note: Significant post-hoc tests of the ANOVA were labeled with numbers (*p* < 0.05).

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
