# Peer review of "Foam Rolling of the Calf and Anterior Thigh: Biomechanical Loads and Acute Effects on Vertical Jump Height and Muscle Stiffness"

_sports, 2019, doi:10.3390/sports7010027_

Round 1

Reviewer 1 Report

Overall a clearly laid out manuscript. The study objective is clear and straightforward and generally the manuscript describes it well. As a result I have only minor comments to make.

line 83 Please confirm both legs received the FR treatment in turn, as it currently reads in such a way that the reader can be confused into thinking one leg is acting as a control.

Figure 3. The y axis currently reads Baseline difference, which is not really true. Perhaps retitle as % relative to baseline?

Author Response

We’d like to thank the reviewer for his comments. Below, we have addressed all comments point by point (A-answer) and used the "Track Changes" function in Microsoft Word, so that all changes are easily visible to the editors and reviewers.

Reviewer 1

Overall a clearly laid out manuscript. The study objective is clear and straightforward and generally the manuscript describes it well. As a result I have only minor comments to make.

A – Thank you for this positive feedback.

line 83 Please confirm both legs received the FR treatment in turn, as it currently reads in such a way that the reader can be confused into thinking one leg is acting as a control.

A – Revised as suggested.

Figure 3. The y axis currently reads Baseline difference, which is not really true. Perhaps retitle as % relative to baseline?

A – Changed as suggested.

Reviewer 2 Report

The authors have evaluated biomechanical loads during different Foam Roller exercises and investigated the acute effects of foam roller on vertical jump height and tissue stiffness. The topic represents a novelty and the manuscripts is also well written. Despite the novelty of the topic I find it difficult to give a practical application to the results.

Below you will find different concerns and suggestions that need to be addressed.

Abstract

Line 24 – 25 It is not clear which forces have been applied and where.

Introduction

Line 37 I would suggest to change “is little” with “has been poorly”.

Line 38 and 39 The sentence reports data collected through force platforms, which measured the pressure of the body on the platform passing through a foam roller. As described it seems that a FR application usually ranges between those two forces reported. I would suggest to reformulate the sentence and make it more clear for the readers and their interpretation.

Line 47 – 48 The authors report different outcomes arising from different studies, and as a consequence (line 49) state that more research is required. Notwithstanding I agree with the authors, I would recommend to add a sentence, in which they briefly state if the studies above mentioned had similar or different assessment protocols or greatly differed in their baseline measures, since these might be crucial factors for the different outcomes reported.

Materials and Methods

Since the authors used a cross over design, was the order of assessment constant or not?

Was there drop-out of participants during the two weeks?

Line 66 Is the comparison between FR and cycle ergometer really necessary to evaluate “biomechanical loads during different FR exercises and to investigate the acute effects on vertical jump height and tissue stiffness”?

Line 73 Did the athletes continue to exercise between the testing sessions? Which kind of recreational activity did the participants perform?

Line 74 I assume this is the case, but I would also suggest to add, since you are trying to evaluate muscular stiffness, to exclude for neurological disabilities.

Line 103 I agree that in order to increase reliability of a measure the mean value should be used, however to evaluate the differences in performance, the maximum value is the correct value to report.

Results

The results for the vertical jump need to be reinterpreted after the maximum value inclusion.

Conclusion

Line 256 I would recommend to add a final sentence in which the authors also report that after 15 minutes the stiffness values returned to baseline.

Thank you.

Author Response

We’d like to thank the reviewer for his time to review our manuscript, which have led to an improved quality. In the following lines, we have addressed all comments point by point (A-answer) and used the "Track Changes" function in Microsoft Word, so that all changes are easily visible to the editors and reviewers.

Reviewer 2

General Comments

The authors have evaluated biomechanical loads during different Foam Roller exercises and investigated the acute effects of foam roller on vertical jump height and tissue stiffness. The topic represents a novelty and the manuscripts is also well written. Despite the novelty of the topic I find it difficult to give a practical application to the results.

Special Comments

Below you will find different concerns and suggestions that need to be addressed.

Abstract

Line 24 – 25 It is not clear which forces have been applied and where.

A – We have reworded this sentence to clarify.

Introduction

Line 37 I would suggest to change “is little” with “has been poorly”.

A – Revised as suggested.

Line 38 and 39 The sentence reports data collected through force platforms, which measured the pressure of the body on the platform passing through a foam roller. As described it seems that a FR application usually ranges between those two forces reported. I would suggest to reformulate the sentence and make it more clear for the readers and their interpretation.

A – We have reworded this sentence to clarify.

Line 47 – 48 The authors report different outcomes arising from different studies, and as a consequence (line 49) state that more research is required. Notwithstanding I agree with the authors, I would recommend to add a sentence, in which they briefly state if the studies above mentioned had similar or different assessment protocols or greatly differed in their baseline measures, since these might be crucial factors for the different outcomes reported.

A – Good point. We have added this topic into the paragraph.

Materials and Methods

Since the authors used a cross over design, was the order of assessment constant or not?

A – As you have asked, a randomized cross-over design was used. Therefore, the order of assessment was not constant. We have already stated this in line 84 and also in figure 1.

Was there drop-out of participants during the two weeks?

A – No, all recruited participants have finished the measurements.

Line 66 Is the comparison between FR and cycle ergometer really necessary to evaluate “biomechanical loads during different FR exercises and to investigate the acute effects on vertical jump height and tissue stiffness”?

A – Good point. No, cycling may not really be necessary. But prior to the study, we have decided to include a further intervention, which may increase the jump performance and therefore may allow a better evaluation of the practical relevance of potentially effects after FR. Cycling was used, because several studies have shown that warm-up procedures, involving ergometer cycling, enhance vertical jump performance.

We have added one sentence to clarify this.

Line 73 Did the athletes continue to exercise between the testing sessions? Which kind of recreational activity did the participants perform?

A – All participants studied sport science and continued their (different) regular sport during the study (e.g. soccer, handball, dancing). Between the testing sessions, the application of FR was not allowed.

We have added one sentence to clarify this.

Line 74 I assume this is the case, but I would also suggest to add, since you are trying to evaluate muscular stiffness, to exclude for neurological disabilities.

A – Thank you. Revised as suggested. We have missed that during writing.

Line 103 I agree that in order to increase reliability of a measure the mean value should be used, however to evaluate the differences in performance, the maximum value is the correct value to report.

A – Thank you for this suggestion, but we disagree with this point. As we want to evaluate the overall effects of FR on the jump performance, the average value may be more reasonable. As for example a subject jumps 3x 30 cm in the first session and 2x 20 cm and 1x 31 cm in the second session, the overall jump performance decrease. Moreover, Al Haddad et al. (2015) have found no differences in the use of the average or best value of three countermovement jumps to monitor changes in the jump performance.

Al Haddad, H., Simpson, B. M., Buchheit, M. (2015). Monitoring changes in jump and sprint performance: best or average values? Int J Sports Physiol Perform, 10(7). 931-934.

Results

The results for the vertical jump need to be reinterpreted after the maximum value inclusion.

A – Please see our comment above.

Conclusion

Line 256 I would recommend to add a final sentence in which the authors also report that after 15 minutes the stiffness values returned to baseline.

A – Corrected as suggested.

Thank you.

Round 2

Reviewer 2 Report

The authors have addressed all the recommended suggestions.

Thank you 

Author Response

Thank you for your time.